# Applying the advocacy coalition framework to wildlife management: Explaining policy change for damage mitigation in Japan

Tatsuya Koga *

Department of Forest Policy and Economics, Forest and Forestry Products Research Institute, Tsukuba, Ibaraki, Japan

☯ The author conducted this research independently.
* koga_tatsuya760@ffpri.go.jp

## Abstract

Case studies in wildlife management and its policy often lack a unified analytical framework, there are few studies that have systematically investigated policy processes and pathways of policy change. This paper examines Japan's ungulate (deer, wild boar) management policy as a case study, applying the Advocacy Coalition Framework (ACF) to the study of wildlife management to elucidate the mechanisms driving policy changes in ungulate management while evaluating the framework's effectiveness. Insights into policy subsystems, policy beliefs, policy-oriented learning, and both external and internal perturbations in the advocacy coalition framework may be useful in understanding the policy change in wildlife management. Data were collected through diet records, official documents, newspaper articles, and semi-structured interviews with stakeholders. Using this data, discourse network analysis was employed to identify coalition structures, resources and policy-oriented learning of two advocacy coalitions—the Hunting Coalition and the Protection Coalition—and events leading to policy change were identified. The analysis revealed that between the 1990s, when ungulate damage became severe, and 2014, the core attributes of Japan's government program transitioned from a centralized protection-focused policy to a decentralized policy promoting hunting. This shift was driven by four pathways proposed in the ACF's bottom-up policy change hypothesis: policy-oriented learning, internal perturbations, external perturbations, and negotiated agreements. These findings highlight the utility of the ACF as an analytical framework. This study suggests that the ACF is a valuable tool for understanding the complex dynamics of wildlife management.

**Data availability statement:** This study includes qualitative interview data involving human research participants, for which strict confidentiality and anonymity were guaranteed as a condition of participation. Access to the underlying data is restricted to protect participant confidentiality and comply with legal constraints. Data requests may be submitted to the research support division of Forestry and Forest Products Research Institute (Email: ronbun@ffpri.go.jp) for evaluation. The corresponding author will not be involved in the decision-making process regarding data access. Requests must include a statement of research purpose and intended use of the data. At the time this study was conducted (in 2024), my institution did not have an ethics review process specifically suited to social science and policy-related interview research involving human participants. A new ethics review process covering such research was introduced at my institution in July 2025. However, this system applies only to research initiated after its implementation and does not retroactively cover earlier studies such as this one.

**Funding:** The author(s) received no specific funding for this work.

**Competing interests:** The Forestry and Forest Products Research Institute (FFPRI), to which I belong, is an organization that appeared in the policy subsystem in Periods 1 and 2, but I declare that there are no known competing financial interests or personal relationships that could affect the research reported in this paper. This does not alter my adherence to PLOS ONE policies on sharing data and materials.

## Introduction

Globally, the overabundance of ungulates has exacerbated a loss of plant diversity, damage to agricultural crops and forestry, ungulate-vehicle collisions, a nuisance to humans, disease transmission to livestock or changes in habitat for other species [1,2]. In particular, since the 1980s, damage caused by ungulates (deer and wild boar) has intensified in Japan [3–6]. To address such damage, between the 1990s and 2014, Japan transitioned from a centrally controlled hunting regulation and wildlife protection policy to a decentralized policy that promotes hunting as a means of mitigating ungulates damage.

Institutional structures, policies, and governance are often path-dependent [7,8]. Several Japanese-language papers have highlighted the strong path dependence in Japan's wildlife management system [9]. In practice, despite the worsening damage caused by ungulates through the early 2000s, policy remained focused on protection. It was not until around 2007 that hunting began to be actively promoted, leading to an increase in the number of ungulates hunted [10]. Why did the previous hunting regulations persist despite the worsening damage, and why was hunting promoted after 2007? Wildlife management is as much a socio-economic-political issue as it is a biological one [11]. By analyzing the politics and policy changes aimed at reducing conflicts between wildlife and human society, and elucidating the factors that facilitated this policy change, this analysis provides valuable insights for future governance and institutional design.

Previous research on wildlife management has not sufficiently examined the pathways through which major policy changes occur [11–13]. Research on policy and governance in wildlife management tends to be ad hoc, making it difficult to find a unified analytical framework. To address these gaps, this paper adopts the Advocacy Coalition Framework (ACF), a widely used framework for analyzing policy processes, to explore the factors that enable policy change in wildlife management [14–16]. The ACF's broad applicability makes it a suitable framework for analyzing wildlife management. The ACF's focus on actors, their belief systems, and scientific findings makes it particularly well-suited for examining wildlife management, where competing values, differing problem views, and the interpretation of scientific evidence are critical [13]. The few existing studies often focus on specific aspects of the ACF. For example, Lauber and Brown [17] emphasize policy-oriented learning, while Nilsson et al. [18] highlight the 'devil shift.' Lundmark et al. [19] offer valuable insights into belief systems and related aspects. Matti and Sandström [20,21] demonstrated that the formation of advocacy coalitions is influenced more by the alignment of policy core beliefs—particularly normative policy beliefs—than by perceived influence. In contrast, deep core beliefs were found to be of relatively minor importance. However, limited empirical analyses comprehensively address key ACF factors, such as policy beliefs, policy-oriented learning, advocacy coalitions, policy subsystems, and external or internal perturbations when examining government program dynamics. Niedziałkowski and Putkowska-Smoter [11], Niedziałkowski et al. [22], and Niedziałkowski [23] developed an original analytical framework by integrating the ACF, evolutionary governance theory, punctuated equilibrium theory, and institutional theory to analyze

the dynamics of wolf management in Poland, Germany, and Belarus, offering numerous insights. By adapting the traditional ACF to the field of wildlife management, this paper aims to extend the methodologies and insights accumulated through ACF research to this domain. Applying the ACF comprehensively to policy areas where it has not been widely utilized allows for testing the framework's adaptability and advancing its development. In this context, this paper aims to analyze the mechanisms that facilitated the policy change toward mitigating wildlife damage and to fill the gap in empirical policy process analysis and ACF studies in wildlife management research. Using the ACF, this paper examines the policy change in Japan's wildlife management and addresses the following two research questions (RQs):

RQ1 Major Policy Change in Wildlife Management: How did Japan's ungulate policy change from its previous focus on protection to emphasizing ungulate damage control? This question seeks to explain the pathway of policy change that led to the promotion of hunting as a response to worsening wildlife damage.

RQ2 ACF in Wildlife Management: Can the ACF be applied to wildlife management studies? To explore this, the paper examines whether the hypotheses developed through previous ACF research are supported by the case under analysis. This examination will also contribute to identifying strategies for future research.

Based on this background, the paper is structured as follows: First, it conducts a review of the literature on wildlife management and the ACF. Next, after briefly outlining the policy change in Japan's ungulates management, it explains the methods and data used for the analysis. Following that, the results of tracing the policy process according to the analytical concepts of the ACF are presented. Based on these results, the paper examines the factors behind the policy change in wildlife management and evaluates the applicability of the ACF to research in this field.

## Theory and framework

Following Partelow's [24] framework usage guide, this section discusses the value of applying the ACF to wildlife management research, the rationale for selecting ACF over other frameworks such as the Multiple Streams Framework (MSF), the data analysis process, and the advantages of empirically testing ACF hypotheses through a wildlife management case study.

### Wildlife management

Since Aldo Leopold's pioneering work in Game Management [25], a substantial body of research has been developed, aiming to both understand and implement the complex dynamics of wildlife management. This field is often described as both an art and a science, involving not only the management of wildlife populations and habitats but also the coordination of human actors for the benefit of wildlife and society [26,27]. Decker et al. [28] further defines wildlife management as "a set of decision-making and implementation activities aimed at influencing several key, interdependent elements: humans, wildlife populations, environments, habitats, and their interactions." Despite the wealth of case studies that have generated valuable insights into the complexities of wildlife management, these studies often remain fragmented and lacking a unified framework. The wide-ranging nature of wildlife management, encompassing diverse topics such as ecology, environment factors, sociology, and policy, makes it difficult to synthesize findings into a coherent theoretical structure.

In order to systematically organize these ad hoc environmental governance case studies and derive theoretical insights, Ohno [29] suggests that "framework thinking" [30] is effective, particularly emphasizing the utility of the ACF. Ostrom distinguishes between frameworks, which provide a meta-theoretical overview across multiple disciplines, and models or theories, which focus on specific hypothesized relationships among variables. A comprehensive meta-theoretical framework like the ACF has the potential to organize the disparate insights drawn from wildlife management, creating a systematic understanding of this complex field. By exploring the applicability and limitations of the ACF in wildlife management, researchers can make meaningful academic contributions. Policy processes and policy changes are central to wildlife management, yet political science and public policy research in this area remain underdeveloped [13]. To unravel these interconnected dynamics, the application of frameworks such as the ACF is essential. However, as this review will show, few studies have systematically applied the ACF to wildlife management, underscoring the need for further research in this

domain. This approach can transform the study of wildlife management from a collection of isolated, ad hoc case studies into a more structured, theory-driven field.

## Advocacy coalition framework (ACF)

**ACF overview.** The Advocacy Coalition Framework (ACF) was developed to provide a systematic understanding of the major factors and processes affecting the overall policy process—including problem definition, policy formulation, implementation, and revision—within a specific policy domain over periods of a decade or more [15]. A distinctive feature of the ACF is its focus on the role of scientific and technical information in the formulation, implementation, and feedback of public policies (Fig 1). To analyze the change in wildlife policy, it seems that the Multiple Streams Framework [31], another well-known policy process analysis framework, could be effective [32]. However, the ACF excels in analyzing policy subsystems over extended periods and focuses on policy-oriented learning informed by scientific evidence. A defining characteristic of the ACF is its adaptability, demonstrated by regular updates that incorporate empirical research applying the framework and theoretical advancements in related fields. ACF was originally developed to analyze policy processes within pluralist political regimes such as that of the United States. However, it has since been refined to enable more comprehensive analysis and is now widely applied to policy process studies across a broad range of countries [15,16,33]. Although Japan, the focus of this analysis, is a non-Western country and not a typical case for ACF application, Ohno et al. [33] argue that applying the ACF to Japan poses no issues. The ACF has been employed in many environmental issues [34–37] and appears effective for analyzing the politics and policies surrounding wildlife management. However, applications of the ACF to the field of wildlife management remain limited [13]. A search of the Web of Science database (conducted on October 11, 2024) brings up only three papers that include both "Advocacy Coalition Framework" and "Wildlife Management" in their title, abstract, or keywords. A review by Sotirov and Memmler [34] suggests that ACF applications in wildlife management studies are less frequent compared to other natural resource policy domains.

In the context of wildlife management, several questions arise: How do actors behave? How do they conflict, compromise, or cooperate with one another? How do administrative wildlife managers make policy decisions amidst competing

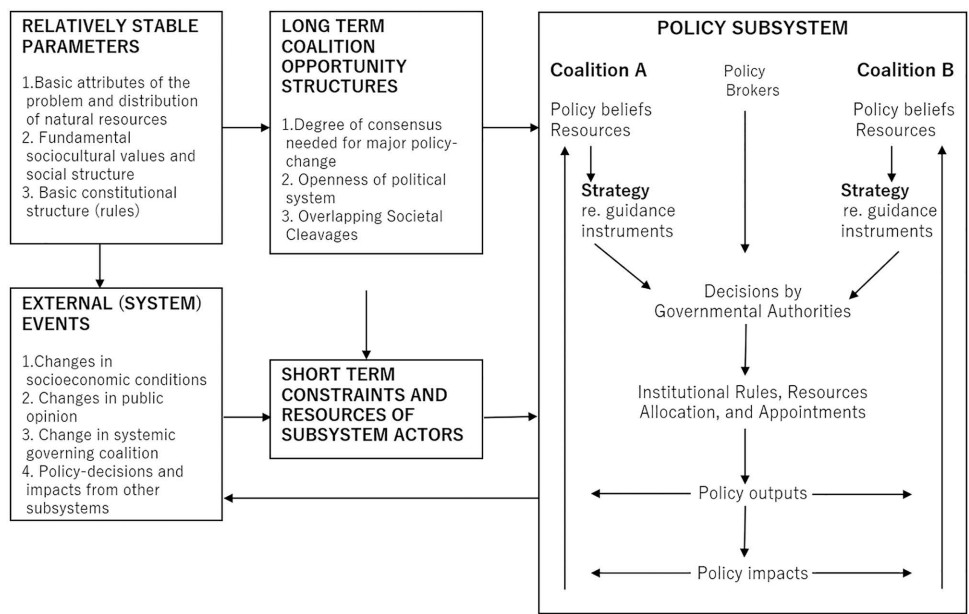

**Fig 1. Advocacy coalition framework.** Source: Prepared by the author based on Nohrstedt et al. [38].

policy beliefs? How do actors learn from scientific findings, advanced examples, and failures? How is wildlife management influenced by external shocks to the policy subsystem? How do core attributes of government programs shift? Why does the status quo persist even when policy change is necessary? Addressing these questions is crucial for advancing both wildlife management research and practice. By using the ACF to analyze wildlife management, systematic insights into these dynamics can be derived.

## Key ACF components

Key components of the ACF include policy subsystems, policy beliefs, advocacy coalitions, bounded rationality, policy brokers, policy-oriented learning, and policy change hypotheses [38].

*Policy Subsystems*: The ACF emphasizes the importance of policy subsystems in the analysis of policy processes, rather than focusing on specific organizations or individuals. A policy subsystem is defined as the primary unit of analysis, determined by geographical areas and substantive topics. It consists of various actors and organizations—such as government agencies, political parties, ministries, NGOs, interest groups, researchers, and the media—that directly or indirectly influence the policy process [39].

*Policy Beliefs*: In the ACF, actors and organizations within a policy subsystem possess a three-layered policy belief system. At the most fundamental level are deep core beliefs, which are normative values that can apply across multiple policy subsystems. Policy core beliefs, situated in the middle of the belief system, are subordinate to deep core beliefs and represent basic policy orientations and value priorities within a specific policy subsystem. The lowest level consists of secondary beliefs, which refer to the policy instruments and methods used to achieve policy objectives [39]. For example, deep core beliefs encompass philosophical views on the appropriate relationship between humans and nature; policy core beliefs pertain to whether hunting should be promoted or restricted; and secondary beliefs involve specific policy instruments or strategies for promoting (or restricting) hunting. While actors in a policy subsystem may alter secondary aspects due to experience, political pressure, or compromise, changes to policy core beliefs are uncommon. Deep core beliefs are generally resistant to change.

*Policy Brokers*: Conflicts often arise between advocacy coalitions within a policy subsystem. Policy brokers, who hold moderate and neutral positions, typically mediate these disputes. These brokers, typically represented by administrative bodies, cabinets, or executive leaders, help mitigate conflicts and facilitate agreements between opposing parties [39].

*Advocacy Coalitions*: Advocacy coalitions within the ACF are groups of actors who share similar beliefs and engage in non-trivial levels of coordination within a policy subsystem. In most cases, the number of advocacy coalitions within a policy subsystem ranges from zero to five, with two being the most common [40]. Understanding a large number of actors as part of a few coalitions makes the complex dynamics of policy subsystems easier to analyze.

*Policy-Oriented Learning*: Actors in a policy subsystem learn policy ideas based on their experiences and the circumstances of opposing advocacy coalitions. Through policy-oriented learning, actors may adjust their policy beliefs within the subsystem [39].

*Bounded Rationality*: The ACF recognizes the existence of cognitive biases in information processing [15]. Due to bounded rationality, actors tend to engage in policy-oriented learning by selectively acquiring information that aligns with their existing policy beliefs [15].

*External Events*: Changes in dynamic socioeconomic conditions, public opinion, systemic governing coalitions, or other policy subsystems can alter the perceptions of actors within a policy subsystem and shift coalition resources, potentially influencing government programs.

*Policy Entrepreneurs*: In some policy subsystems, innovative actors known as policy entrepreneurs strategically deploy their political resources by proposing policy ideas and building coalitions at opportune moments [41,42]. These key individuals often play a pivotal role in expanding coalition resources, promoting policy-oriented learning, and facilitating policy change [43].

### ACF and wildlife management

Among the 12 hypotheses presented by the ACF (S1 Table), those addressing policy change are particularly relevant to wildlife management. Testing these hypotheses provides insights into actors' policy beliefs, coalition dynamics, policy-oriented learning, and the mechanisms driving policy change in this domain. This study empirically examines policy changes in wildlife management (RQ1) and evaluates the potential of the ACF to advance the field (RQ2), with a specific focus on policy change hypotheses.

Policy Change Hypothesis 1 (PCH1) proposes mechanisms for "bottom-up policy change" through four pathways: external perturbations, internal perturbations, policy-oriented learning, and negotiated agreements [44]. Policy Change Hypothesis 2 (PCH2) addresses "top-down policy change," hypothesizing the role of higher judicial authority in these processes [44]. These two hypotheses are expected to provide valuable insights into the pathways and characteristics of policy change, especially in the context of Japan's ungulate policies, which are examined in this paper.

## Case and methods

### Case

**Overview of the case.** Japan's wildlife management policies were historically centralized, with the Forestry Agency (FA) and the Ministry of the Environment (ME) regulating hunting. This centralized control dates back to the comprehensive revision of the Hunting Act in 1918 and the subsequent amendment of the Wildlife Protection and Hunting Act (WPHA) in 1963 [45,46]. Prior to 1999, regulations were strict: hunters were allowed to hunt no more than one male deer per day, and hunting female deer was strictly prohibited. The focus was on wildlife protection through stringent hunting regulations and the designation of wildlife protection areas.

In Japan, wildlife hunting is categorized into two main types: recreational hunting and certificated culling authorized by governmental bodies for wildlife damage control purposes. Both types are predominantly conducted by local hunters. However, there is a significant distinction in terms of the financial arrangements associated with each type. In the case of recreational hunting, hunters are required to pay a hunting tax in order to participate. On the other hand, for certificated culling, which is conducted to mitigate wildlife damage, local hunters are compensated financially by governmental authorities, such as prefectural or municipal administrations [5,45].

Damage caused by ungulates—particularly deer—worsened in the 1980s, prompting the need for policy change. Pioneering prefectures such as Hokkaido and Iwate began implementing scientific monitoring and population management programs in response to the increasing damage in the 1990s. By the late 1990s, the ruling Liberal Democratic Party (LDP) and the ME pushed for a more decentralized system, institutionalizing a new wildlife management approach that emphasized reducing ungulates damage. However, this push for decentralization was not without opposition. Environmental NGOs, the Democratic Party of Japan (DPJ), the Japanese Communist Party (JCP), and the Social Democratic Party (SDP), and other advocacy groups criticized the promotion of hunting and the relaxation of regulations, advocating instead for continued national-level control and stronger wildlife protections. Despite these conflicts, the 1999 revision of the WPHA passed, marking a significant shift toward the decentralization of wildlife management to prefectures.

After the 1999 revision, the ME continued its decentralization efforts, culminating in a policy change in 2007: the nationwide ban on hunting female deer was lifted. Throughout the 2000s, wild boar damage emerged as a significant issue alongside deer damage. By 2007, all political parties, including former opponents, endorsed the LDP's proposal for the Act on Special Measures for the Prevention of Damage Related to Agriculture, Forestry, and Fisheries Caused by Wildlife (ASMPDRAFFCW). This law further decentralized authority to municipalities and increased funding for wildlife damage prevention measures through the MAFF (Ministry of Agriculture, Forestry and Fisheries). As Fig 2 shows, this led to a significant increase in certificated culling numbers.

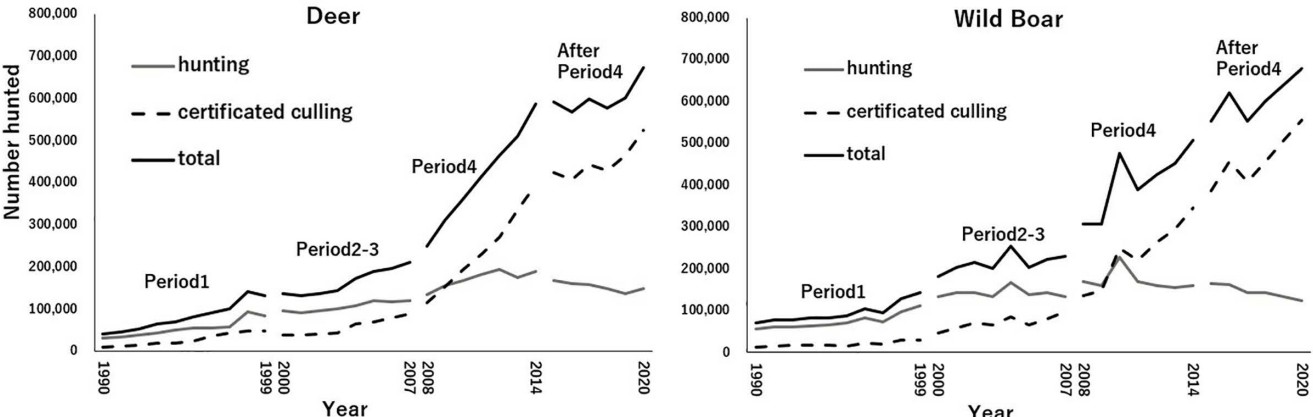

**Fig 2. The Number of Hunted Deer and Wild Boar during 1990-2020.** Source: Prepared by the author based on data from the Ministry of the Environment.

In 2013, the ME and the MAFF jointly set a target to halve the populations of deer and wild boars. The following year, in 2014, the WPHA was significantly amended and renamed the Wildlife Protection, Control, and Hunting Management Act (WPCHMA). This amendment marked a departure from the protective policies that had dominated the past century, transforming hunting from a regulated activity into a public project focused on proactive ungulate population control. With this change, the emphasis firmly shifted toward managing ungulates as a means to mitigate damage, rather than merely protecting species through restrictive regulations.

**Summary of key policy changes.** Japan's wildlife management policies underwent a major transformation from the 1990s to the 2014 WPCHMA amendment. This transformation can be summarized as a shift from centralized wildlife protection, which focused on regulation of hunting under national control, to decentralized management and the promotion of hunting to prevent damage caused by ungulates. Certificated culling increased significantly as the central focus moved toward controlling ungulate populations to protect agricultural and forestry interests. The analysis of this policy change is divided into four key periods:

Period 1 (up to 1999): This period covers the lead-up to the 1999 WPHA revision, which introduced decentralization to prefectures in response to increasing deer damage and the establishment of an ungulates management plans system.

Period 2 (2000–November 2007): Characterized by the relaxation of hunting regulations, including the gradual easing of restrictions on hunting.

Period 3 (November 2007): Marked by the enactment of the ASMPDRAFFCW, which further decentralized authority to municipalities and expanded financial resources for wildlife damage prevention measures.

Period 4 (2009–2014): During this period, ME and MAFF set targets to halve deer and wild boar populations, culminating in the 2014 WPCHMA revision, which transformed hunting into a public project for proactive population control.

## Data collection and analysis

**Data collection.** I collected various sources including newspaper articles (Asahi Shimbun), meeting transcripts (Diet records and committee meeting minutes), party materials (such as Monthly Liberal Democrat, Parliament and Local Governments, Komei Shimbun, Shimbun Akahata, and Policy Special Report), industry papers, academic literature, and interview records and autobiographies of relevant individuals. Additionally, I made formal requests to the government for public records.

As a supplement, I conducted semi-structured interviews with 25 people involved in Japan's ungulate policy, and obtained information that had not been documented or confirmed as factual (S2 Table). Due to factors such as the death

of relevant individuals and the dissolution of certain organizations, it was not feasible to interview all entities listed in S3 Table. Nevertheless, semi-structured interviews were conducted with 25 individuals who had in-depth knowledge of the policy process at the time. Interviews focused on actors' perceptions of policy change, engagement with scientific data, and interactions with opposing coalitions. Specific questions addressed how participants processed new scientific information, whether their policy beliefs shifted over time, and what factors influenced their stance on hunting promotion (S2 Table). Ethics approval was not obtained for this study, as it was not required under the guidelines of my institution. The research involved a qualitative case study consisting of non-sensitive, semi-structured interviews with adult professionals (e.g., researchers, policymakers) on their institutional and policy experiences, and analysis of publicly available documents. No personal, medical, or confidential information was collected, and all participants were fully informed of the research purpose and gave verbal consent prior to participation. This consent was audio-recorded at the beginning of each interview. There was no foreseeable risk to participants. In accordance with my institution's ethical standards for social science research, such a study does not require formal ethics review. To demonstrate that these interviews do not pose any ethical concerns, an overview of the interview procedures is provided in Supporting Information (S2 Table).

**Analytical procedure and coding scheme.** Actors within the policy subsystem were identified based on the following criteria: (1) actors who expressed policy preferences in the Diet or advisory committees during Periods 1–4, (2) actors whose influence on policy was confirmed through interviews, and (3) actors who submitted petitions or statements to the Diet or the ME. Based on these criteria, I first compiled a list of actors within the policy subsystem in each period. This list was reviewed by actors involved in the policy subsystem, and minor adjustments were made (S3 Table).

I analyzed the coalition structure and resources using discourse network analysis (DNA) based on publicly available materials [47]. The codebook was developed based on the examples of policy core beliefs provided by Sabatier [39], specifically focusing on issues that frequently arose in debates: orientation on basic value priorities, identification of groups or other entities whose welfare is of greatest concern, and proper distribution of authority among levels of government. To analyze secondary beliefs, I also coded whether actors agreed or disagreed with major legislation related to ungulate management, as well as their stance on the lifting of the hunting ban on female deer and the simplification of obtaining a trapping license (S4 Table). Since not enough deep core beliefs were observed, these were excluded from the coding process. To enhance the accuracy of the coding process, a hired graduate student specializing in Japanese wildlife management independently reviewed the coding results. In cases where discrepancies arose between our coding outcomes, I engaged in discussions with her to reach a consensus on the final coding decisions. A total of 102 documents were used for the DNA analysis (32 newspaper articles, 29 diet records, 37 petitions to the government, and 4 other documents), with a total of 443 policy concepts coded across various agencies. This process revealed two coalitions: the protection coalition, which seeks to protect ungulates by maintaining hunting regulations, and the hunting coalition, which seeks to reduce populations through the promotion of hunting.

To identify coalition resources, I referenced Nohrstedt [48] and clarified, through the collected documents and interviews, how each coalition utilized public opinion, information, leadership, and legal authority during each period. Additionally, I examined whether events outside the policy subsystem influenced ungulate policy by analyzing the collected documents and interviews. For the analysis of qualitative data, the software MAXQDA was employed.

## Results

### Period 1 (~1999): The intensification of deer damage, policy-oriented learning, and decentralization triggered by external perturbations

**Intensifying deer damage.** Since the comprehensive revision of the Hunting Act in 1918, Japan's wildlife management had long been defined by centralized hunting regulations and wildlife protection policies aimed at increasing wildlife populations through feeding and protection measures [46,49]. Deer and wild boar were not subject to systematic population control. While localized agricultural and forestry damage caused by ungulates occurred sporadically, it had

not yet become a national concern. Deer populations were relatively sparse, and issues such as the loss of understory vegetation were not widespread.

In 1979, a group of mammal researchers conducted Japan's first nationwide survey of mammal distribution. The report indicated that deer populations were fragmented and declining in most regions, leading to calls for enhanced protection [50]. This study significantly influenced researchers and forestry officials, reinforcing the perceived importance of hunting regulations and habitat conservation to protect species. During the 1980s, as deer populations and their habitats began to recover, forestry damage increased. Nevertheless, a strong belief persisted among researchers and forestry officials that maintaining deer protection was vital. One interviewee recalled that a "normalcy bias" delayed recognition of the expanding deer population and the need for damage control measures. Other interviewees responded as follows: "Everyone thought deer damage would eventually subside". One interviewee stated, "Everyone thought they were on the brink of extinction or mythical creatures. [...] While evidence is said to be critical in wildlife management, there wasn't any evidence to begin with. By the time we waited for evidence, the population had grown out of control."

**Coalition structures and policy-oriented learning.** By the early 1980s in Hokkaido Prefecture and the late 1980s in other regions, the impact of deer damage on agriculture and forestry became more apparent. Members of the hunting coalition—including researchers, prefectural officials, forestry officials, and ME staff—argued that factors such as mild winters, clear-cutting practices, previous protection policies, and a decline in hunter numbers contributed to the growing deer populations. They emphasized the importance of promoting hunting and installing protective fencing, with particular focus on population control through hunting.

Initially, non-lethal methods like repellents and fences were preferred. However, researchers learned that these approaches were insufficient, as deer would simply relocate and cause damage in other areas. The high costs associated with extensive fencing and skepticism regarding the efficacy of repellents led to increasing advocacy for hunting as a control measure. The ruling LDP, closely aligned with agricultural interests, established an intra-party group dedicated to addressing agricultural damage and published a report in 1997 outlining wildlife and hunting system reforms that advanced damage control initiatives.

Conversely, the protection coalition opposed these measures, asserting that existing protection policies were not overly restrictive, and attributing the increase in deer populations to factors such as mild winters and deforestation. They argued that the scientific data on deer at the time was uncertain and predicted a future population decline that would alleviate the damage. For instance, a 1991 report by the Nature Conservation Society of Japan (NACS-J) emphasized that the population increase was temporary and opposed drastic measures such as hunting. The protection coalition advocated for non-lethal strategies and maintaining current hunting regulations.

Coalition resources, including leadership, played a critical role in the protection coalition. Policy entrepreneurs within the protection coalition built alliances with left-wing opposition parties, such as the Democratic Party of Japan (DPJ), the Japanese Communist Party (JCP), and the Social Democratic Party (SDP). These policy entrepreneurs also established the Wildlife Protection Act Network (WPAN), which actively campaigned against lethal measures and decentralization (Fig 3).

**External perturbations and policy change.** The policy change in Period 1 was the 1999 WPHA revision, which decentralized wildlife management authority to the prefectures, driven by external perturbations. In the 1990s, nationwide efforts toward decentralization prompted the ME to encourage prefectures to develop wildlife management plans, particularly for deer. To mitigate damage caused by deer, policy entrepreneurs like Shingo Miura (Forestry and Forest Products Research Institute) and Koichi Kaji (Hokkaido Environmental Research Center) in the Iwate prefecture and Hokkaido prefecture actively promoted hunting and developed their own deer management plans, which served as a tailwind for these efforts [51,52]. They proposed concepts such as "adaptive management" for wildlife management under uncertainty and lack of scientific information, and incorporated these concepts into their own deer management practices. Prefectures that adopted such management plans were allowed to lift restrictions on deer hunting, including the hunting of female deer. However, actors within the hunting coalition considered the significant impact that lifting the hunting ban

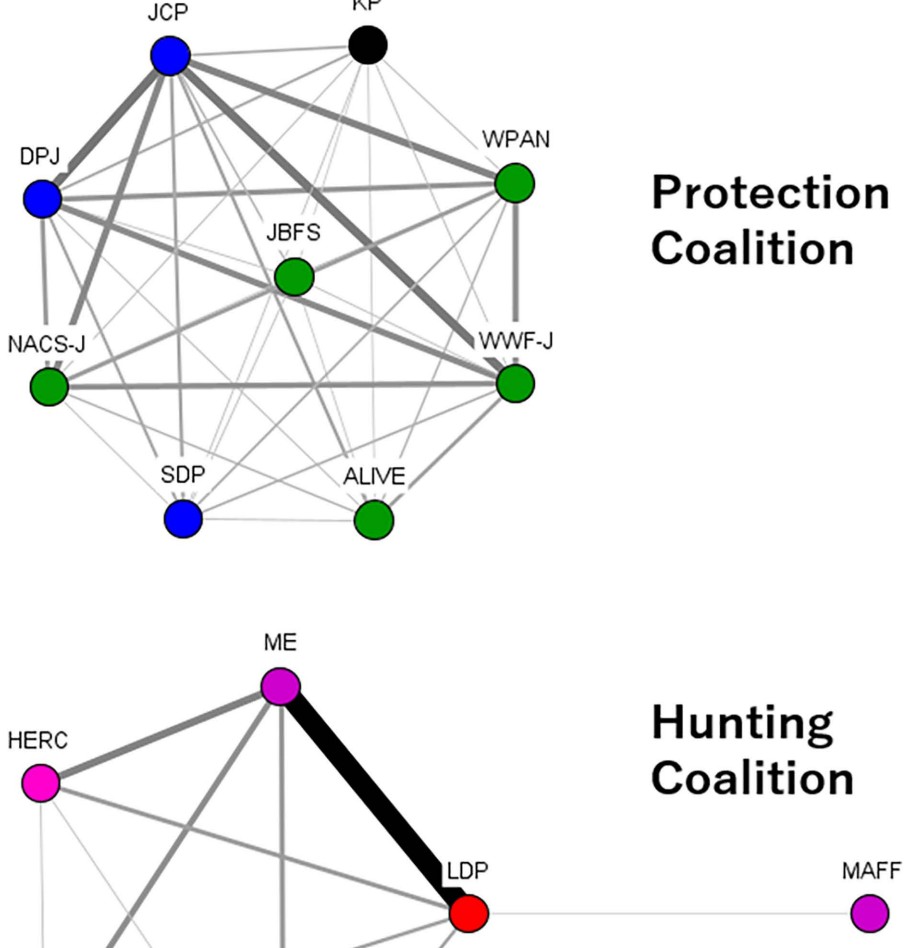

**Fig 3. DNA in Period 1.** Note: The color of the node indicates each organizational attribute. Red indicates right-wing parties, blue indicates left-wing parties, green indicates nature conservation groups and animal protection groups, pink indicates research institutes, orange indicates hunting organizations, purple indicates governments. Source: The Author.

on female deer could have on population control. As a result, they opted for a cautious approach, advocating for lifting the ban only in regions with established management plans, rather than implementing a nationwide policy. Additionally, there were differing opinions within the hunting coalition itself; some actors argued that promoting the hunting of male deer alone would suffice as a response to the growing ungulate population. An interviewee within the hunting coalition stated, "I understood the necessity of promoting hunting, but honestly, I was opposed to lifting the hunting ban on female deer. I also expected their numbers to return to near extinction. [...] Looking back now, if the ban on hunting female deer had been lifted 10 years earlier, the deer damage might not have become this severe."

The protection coalition strongly opposed these measures, arguing that prefectures lacked adequate scientific knowledge and monitoring systems. Despite this opposition, the ruling LDP successfully pushed the legislation through. The passage of the bill was reportedly a high-stakes matter for senior ME officials, who faced significant pressure to persuade NGOs and opposition parties. The Komeito party (KP) eventually supported the bill following negotiations. However, two additional provisions were added as compromises: (1) a review of the law's implementation in 2002 to prevent overhunting and (2) eight supplementary resolutions to safeguard against excessive hunting. The protection coalition remained opposed to the bill, expressing concerns that it would lead to overhunting and potentially cause species extinction.

In Period 1, the core policy attribute—ungulate protection—remained unchanged.

### Period 2 (2000–2007.11): policy-oriented learning of the hunting coalition and the lifting of the ban on female deer hunting

**Policy feedback of decentralization in period 1, coalition structure and policy-oriented learning.** Prior to the 1999 WPHA revision, wildlife protection projects were carried out by prefectures under orders from the national government. After the revision, prefectures gained autonomy over ungulate management, enabling them to lift hunting bans and ease regulations related to hunting tools and seasons. However, this transfer of authority was not accompanied by financial support, leading many prefectures to respond by loosening hunting regulations. Although some prefectures attempted to provide subsidies for hunting, these measures were limited in scope.

While the primary concern during Period 1 was the increase in deer populations, Period 2 saw growing recognition of the escalating damage caused by wild boars. In parallel, the decentralization initiated in Period 1 set the stage for innovative approaches to ungulate management at the local level. For example, Shimane Prefecture simplified the process for obtaining trap hunting licenses and relaxed regulations on trap hunting, resulting in a notable increase in wild boar hunting. The hunting coalition learned from such cases, recognizing the importance of promoting trap hunting as a strategy for agricultural damage control (policy-oriented learning). However, the protection coalition opposed the expansion of trap hunting (Fig 4).

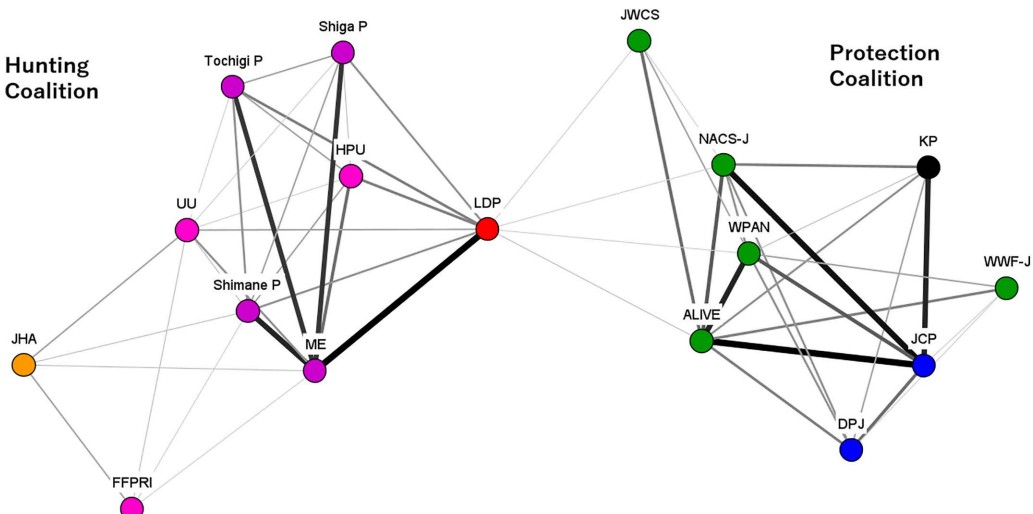

**Fig 4. DNA in Period 2.** Note: The color of the node indicates each organizational attribute. Red indicates right-wing parties, blue indicates left-wing parties, green indicates nature conservation groups and animal protection groups, pink indicates research institutes, orange indicates hunting organizations, purple indicates governments. Source: The Author.

During the 2006 WPHA revision and throughout Period 2, however the core policy beliefs of the protection coalition remained unchanged. However, the secondary aspects of the left-wing parties' stance showed some softening. They acknowledged the necessity of deer and wild boar hunting in unavoidable circumstances, although they continued to voice concerns about the active promotion of hunting. They emphasized that habitat conservation and protective fencing should be prioritized, with hunting for population control only to be considered when these measures failed to prevent damage.

**Policy change towards relaxing of hunting regulations.** During the 2006 WPHA revision, the ME, the LDP, and prefectural governments sought to further promote trap hunting by simplifying the process for obtaining trap hunting licenses. Despite opposition from left-wing parties within the protection coalition, the bill passed due to the LDP's majority in both houses of the National Diet. Additionally, in the prefectures, which had become the main actors in ungulates management, there was a growing recognition that hunting ungulates was necessary even in temporarily closed hunting areas. Some researchers within the hunting coalition believed that the ban on hunting in these areas had contributed to the increase in ungulate populations. Consequently, the hunting coalition advocated for the lifting of hunting restrictions in temporarily closed areas, which was institutionalized through the 2006 WPHA revision. One interviewee described this period as "a time when motivated prefectures had been asking the national government to relax regulations that would be barriers to hunting promotion". Moreover, many prefectures began to lift the ban on hunting female deer, a trend that ultimately led to the adoption of a nationwide policy in 2007.

In Period 2, while hunting regulations were relaxed, the core policy attribute—ungulate protection—remained unchanged.

### Period 3 (2007.12): Venue shopping and policy change

**Venue shopping.** Since 2007, the LDP began drafting a bill for the ASMPDRAFFCW, focusing on three key points: (1) utilizing the MAFF's budget for wildlife damage control, (2) transferring some authority to municipalities and expanding their financial resources, and (3) involving the Self-Defense Forces (SDF) in hunting. The LDP believed that, to effectively prevent damage to agriculture and forestry caused by ungulates, it was necessary to establish a law under the MAFF that was separate from the WPHA under the jurisdiction of the ME and specifically addressed wildlife damage measures. Additionally, the LDP emphasized the importance of decentralization and providing financial resources to municipalities, which were closer to the sites of damage than prefectural governments.

Although some actors within the protection coalition opposed the bill, the most contentious issue was point (3) the involvement of the SDF in hunting. Left-wing parties and nature protection groups within the protection coalition viewed the existence of the SDF as problematic, with many advocating for its abolition. Some actors within the protection coalition, particularly nature protection groups and animal welfare organizations, launched a campaign against the ASMP-DRAFFCW. Ultimately, the LDP abandoned point (3)—the involvement of the SDF—but submitted the bill with (1) the utilization of MAFF's budget for wildlife damage control and (2) the transfer of some authority to municipalities. The bill was passed unanimously.

**Coalition structure and policy change.** Notably, even within the protection coalition, parties like the DPJ supported the bill, acknowledging that while a long-term shift to non-lethal methods was necessary, promoting hunting was essential in the short term (Fig 5). During Periods 1 and 2, both the DPJ and the JCP had opposed hunting as a means of ungulate damage control, with some advocating for a complete ban on hunting. However, by Period 3, they supported this bill that promoted hunting. One interviewee (neutral position) noted, "They (the left-wing parties) weren't exactly in favor of actively promoting hunting, but the wildlife damage had become so severe that they simply couldn't oppose it. There was an atmosphere that if they (the left-wing parties) didn't support this bill, they wouldn't get the support of those in the agriculture and forestry industries who were suffering damage." More than half of the interviewees agreed that, during Period 3, the positioning of ungulate management as part of agricultural and forestry policy made it more difficult for political parties to oppose hunting promotion initiatives.

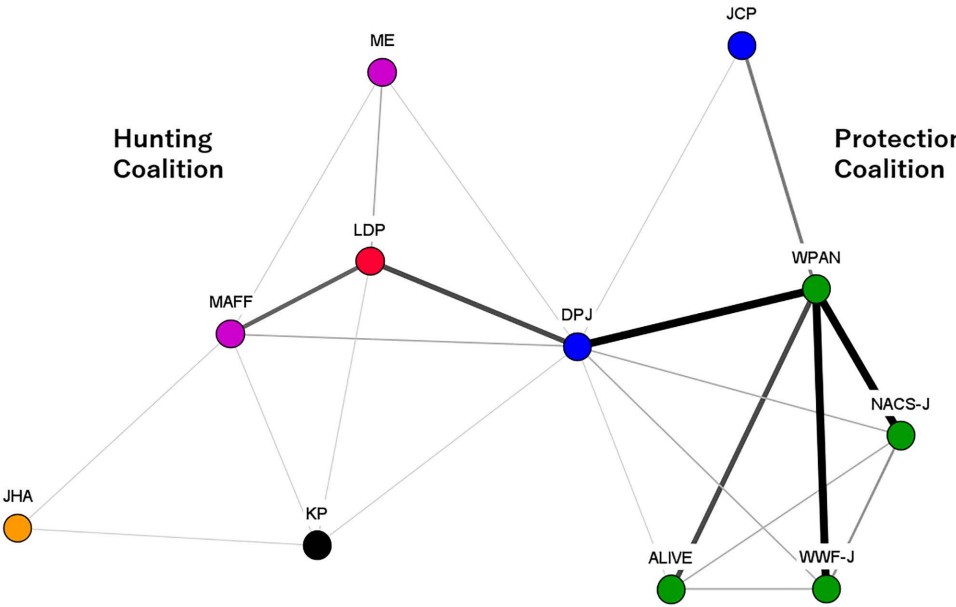

**Fig 5. DNA in Period 3.** Note: The color of the node indicates each organizational attribute. Red indicates right-wing parties, blue indicates left-wing parties, green indicates nature conservation groups and animal protection groups, pink indicates research institutes, orange indicates hunting organizations, purple indicates governments. Source: The Author.

In fact, the DPJ cooperated with the LDP during the drafting stage of the legislation, though they continued to express concerns about overhunting. The JCP also supported the bill but maintained concerns about the potential for overhunting as a result of promoting hunting. This shift suggests that voting for the bill of left-wing parties occurred in response to public pressure rather than scientific data showing that wildlife damage was becoming more serious, with ASMPDRAFFCW's promotion of hunting as a focal point.

As a result of the ASMPDRAFFCW's passage, the initiative for hunting management shifted further from the national government to municipalities, whose authority was strengthened. While the decentralization in Period 1 transferred authority to prefectures without financial support, Period 3 saw municipalities receiving financial backing from the MAFF, facilitating the provision of hunting subsidies. Tokita [10] highlights that while the policy of relaxing hunting regulations before the establishment of the ASMPDRAFFCW in 2007 did not lead to a significant increase in the number of hunted ungulates, the law's passage in 2007 led to subsidies being provided for hunting (particularly certified hunting), resulting in a significant increase in the number of hunted ungulates (Fig 2).

Period 3 resulted in venue shopping [53], which positioned wildlife management as a MAFF agroforestry policy issue rather than a Ministry of Environment policy, and the core attributes moved toward mitigating ungulate damage by promoting hunting.

### Period4 (2008–2014): Policy change toward mitigating damage caused by ungulates

**Change in government.** From Period 3 onwards, the severity of damage caused by deer and wild boars became increasingly apparent, leading even left-wing parties within the protection coalition to eventually accept the promotion of hunting. Nature conservation Group also came to recognize the necessity of promoting hunting, as they became increasingly concerned about the decline of understory vegetation and the loss of biodiversity caused by deer herbivory. Consequently, national-level conflicts over the eradication of deer and wild boars subsided.

During Periods 1 and 2, the DPJ, a member of the protection coalition, held power from 2009 to 2012. However, the DPJ did not propose significant protective policies. The LDP, then in the opposition, proposed an amendment to the ASMPDRAFFCW in 2012 to address the escalating damage caused by ungulates. Prime Minister Yoshihiko Noda (2011–2012) of the DPJ also supported the promotion of hunting as a means to mitigate ungulate damage.

**Policy-oriented learning.** In 2013, the first nationwide estimate of the deer population was performed during a meeting of experts (The Committee on Wildlife Protection and Management). According to the estimate, the deer population had been rapidly increasing since the 1990s. If the hunting rate were maintained, the population was projected to double within 10 years. Conversely, doubling the hunting rate could halve the population over the same period. This simulation had a significant impact on government actors, highlighting the urgent need to promote hunting (policy-oriented learning). With growing calls for the ME to set population reduction targets, MAFF announced a policy to double the deer hunting rate and reduce the population by half. Simultaneously, it was decided to halve the wild boar population.

To achieve the goal of halving the populations of deer and wild boar, the ME sought to amend the WPHA in 2014. This amendment represented a major shift in Japan's WPHA, which since the 1918 revision had primarily focused on wildlife protection. The amendment aimed to mitigate damage caused by ungulates and manage populations through hunting. The revised law, now titled the Wildlife Protection, Control, and Hunting Management Act (WPCHMA), incorporated three main features.

1. The emphasis shifted from "wildlife protection through hunting regulations" to "population management."

2. A system was established to designate certain wildlife species causing severe damage as "Wildlife Species Designated for Management," with deer and wild boar being the initial species under this system.

3. A program was created to enable national and prefectural governments to conduct hunting as public projects (Programs of Capturing Wildlife Species Designated for Management), allowing private culling organizations to participate, in addition to traditional local hunting organizations.

The ruling LDP, which had strong support in rural areas, prioritized reducing wildlife damage and backed this bill. The DPJ, which had become the main opposition after losing power in 2012, had opposed the promotion of hunting during Periods 1 and 2 but showed some support for this bill (Fig 6).

**Coalition structure and policy change.** The issue of conducting public project hunting as a government initiative elicited diverse opinions. Some prominent wildlife management researchers argued that sustaining hunting with the aging and shrinking population of general hunters was challenging and recommended that the government employ professional hunters as public servants. Conversely, the Japan Hunting Association (JHA), a national hunters' association, opposed this due to concerns over reduced hunting opportunities and competition from private culling organizations. WWF-Japan also opposed both the public management of hunting and the inclusion of private organizations. However, as these conflicts occurred at the secondary belief level, they did not escalate into major issues. The bill was ultimately passed in the National Diet, establishing hunting as a public project to mitigate damage caused by ungulates.

Following Period 4, the promotion of hunting led to a significant increase in hunting activities (Fig 2), resulting in a downward trend in the populations of deer and wild boar. Nonetheless, challenges remain, including the decreasing and aging population of hunters, who are essential for ungulate management [2], low resource utilization of hunted deer and wild boar [54], and managing ungulate populations at Self-Defense Forces bases, which restrict entry [55].

## Discussion

### RQ1: Major policy change in wildlife management

Referring to PCH1, I will explain the pathways leading to policy change in Japan's ungulate management. Despite the increasing damage, policy changes during Period 1 were limited to decentralization and the partial lifting of the hunting ban on female deer. Several factors constrained policy change toward promoting hunting.

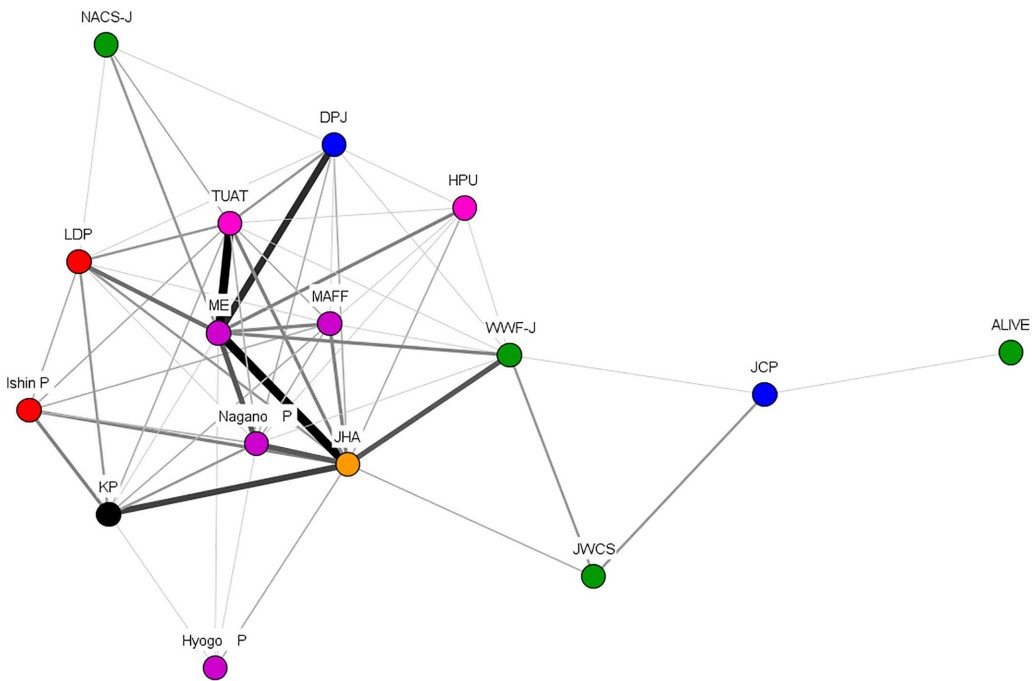

**Fig 6. DNA in Period 4.** Note: The color of the node indicates each organizational attribute. Red indicates right-wing parties, blue indicates left-wing parties, green indicates nature conservation groups and animal protection groups, pink indicates research institutes, orange indicates hunting organizations, purple indicates governments. Source: The Author.

1. Cognitive biases regarding ungulate damage and the increase in ungulate populations, as well as the resulting limitations on policy-oriented learning: In Period 1, the view of ungulates as "near extinction" or "mythical animals" persisted even as populations grew, limiting the resources available to the hunting coalition. For instance, the protection coalition suggested that the increase in deer populations was temporary and would soon decline. Additionally, hesitation within the hunting coalition regarding a national-level ban lift on female deer hunting stemmed from concerns about population control. This cautious approach was influenced by what one interviewee referred to as a "normalcy bias." Also, the statement by one interviewee, "While evidence is said to be critical in wildlife management, there wasn't any evidence to begin with," aptly illustrates the absence of policy-oriented learning during Period 1. This lack of empirical data and evidence hindered informed decision-making and delayed the recognition of the pressing need for policy change in ungulate management. The ACF suggests that policy-oriented learning can be hindered by cognitive biases preventing actors from adapting to new realities [15].

2. Opposition from the Protection Coalition: Policy entrepreneurs within the protection coalition maintained significant resources to oppose hunting promotion. Throughout Period 1, the coalition criticized hunting, citing scientific uncertainty. As a result, policy changes were limited to decentralization while maintaining the core attribute of ungulate protection. Coalition building and networking, strategies known to be used by policy entrepreneurs [42], were evident as the protection coalition created the WPAN to support their anti-hunting promotion campaign, which persisted until Period 3.

Factors enabling policy change from Period 3 onward included the escalation of ungulate damage (internal perturbation), the expansion of coalition resources via policy-oriented learning, and the decentralization initiated in Period 1, which expanded the hunting coalition's resources. These shifts can be analyzed through coalition resources such as leadership, legal authority, information, and public opinion [48].

The decentralization initiated in Period 1 had significant impacts on the political dynamics of Period 2 and beyond. The 1999 transfer of authority from the national government to prefectures allowed proactive prefectures to lift the ban on female deer hunting. Shimane Prefecture's innovative relaxation of trap hunting licenses influenced national policy, culminating in the nationwide ban lift in 2007. Requests from prefectural officials drove the decision to lift hunting bans in closed hunting areas during Period 2.

Leadership was pivotal as a coalition resource. Policy entrepreneurs like Koichi Kaji and Shingo Miura from the hunting coalition advocated for proactive hunting promotion policies to mitigate damage. Their efforts led to initiatives such as the lifting of the female deer hunting ban, adaptive management, and the development of population management plans at the prefectural level. Such policy shifts, facilitated by demonstration strategies of the policy entrepreneur [42], are known to spread policy ideas that enable change [41]. The importance of the policy entrepreneur has been noted in wildlife management policy studies [11,56].

The Legal Authority played a critical role. The expansion of legal authority at the prefectural level in Period 1 laid the groundwork for subsequent promotion of hunting. Drawing on policy feedback theory [57], Schmid et al. [58] have linked it to the ACF, suggesting that policies can influence coalition resources and actor emergence. In this view, the resource expansion of the hunting coalition in Period 2 can be seen as an outcome of policies from Period 1. Advanced prefectures like Iwate, Hyogo, Hokkaido, Nagano, Shimane, and Tochigi served as models for others and influenced national policy. Officials and researchers from these regions were invited to the ME and the Diet, promoting hunting to mitigate damage and proposing policies, facilitating policy-oriented learning.

Public opinion in Period 3, particularly from those affected by agricultural and forestry damage, strongly supported hunting promotion, making it difficult for left-wing parties to oppose. Information, particularly scientific data showing rapid population growth, had a notable impact in Period 4, enhancing support for hunting promotion among left-wing parties and environment protection organizations.

Over the four periods, all four pathways for policy change proposed by the ACF were evident. In Period 1, external perturbation (decentralization) occurred, while negotiations between the KP and the ME resulted in decentralization. Internal perturbations (wildlife damage) persisted throughout Periods 1–4. Policy-oriented learning became increasingly important throughout Periods 1–4. These findings support the ACF's first hypothesis (PCH1), emphasizing pathways that explain policy change.

Governmental changes in 2009 and 2012 had minimal impact on ungulate management. This was due to the DPJ adopting a more hunting-friendly stance in Period 4. Pierce's classification of policy shifts—bottom-up changes driven by perturbations and policy-oriented learning, versus top-down changes from jurisdictional shifts—supports the view that Japan's shift in ungulate management was a bottom-up policy change (PCH1) rather than top-down (PCH2).

**RQ2: ACF in wildlife management studies**

This study applied the Advocacy Coalition Framework (ACF) to analyze Japan's wildlife management policies, providing valuable insights into the dynamics of advocacy coalitions, external perturbations, policy-oriented learning, and policy decisions. As shown in Table 1, the validation of many ACF hypotheses demonstrates its effectiveness as an analytical tool for wildlife management research. Out of 12 hypotheses, 9 were either fully or partially supported. However, the findings point to two significant considerations regarding the ACF's hypotheses, particularly regarding the roles of venue shopping and policy brokers.

First, during Period 3, parties such as KP and left-wing parties (DPJ, JCP) that had previously aligned with the protection coalition supported bills promoting hunting, it was driven by a reluctance to oppose hunting promotion, motivated by venue shopping. As one interviewee noted, they "couldn't oppose it." This shift was primarily influenced by the interests of the agricultural and forestry sectors. As indicated by the DNA's positioning, left-wing political parties, JCP had aligned with the protection coalition, and the DPJ had positioned a neutral stance in Period3 (Fig 5). However, they too eventually

Table 1. Testing the ACF hypothesis.

| Hypothesis | Hypotheses and their validation results |
|---|---|
| **Coalition Hypothesis 1 (CH1)** | On major controversies within a policy subsystem when policy core beliefs are in dispute, the lineup of allies and opponents tends to be rather stable over periods of a decade or so. |
| | **2**: (e.g.,) Although there were changes in policy core beliefs after Period 3, left-wing parties and environmental protection groups opposed the promotion of hunting up to Period 2, while the LDP, the Ministry of the Environment, and the Japan Hunting Association tried to promote hunting from Period 1–4. |
| **Coalition Hypothesis 2 (CH2)** | Actors within an advocacy coalition will show substantial consensus on issues pertaining to the policy core, although less so on secondary aspects. |
| | **2**: (e.g.,) In Period 4, of the actors that belonged to the hunting coalition, the Ministry of the Environment and some researchers tried to allow private culling organizations to participate, but the JHA tried to promote hunting on its own. |
| **Coalition Hypothesis 3 (CH3)** | Actors (or coalitions) will give up the secondary aspects of their belief systems before acknowledging weaknesses in the policy core. |
| | **2**: (e.g.,) In Period 3, in exchange for giving up on the LDP's participation in hunting by the Self-Defense Forces (secondary aspects), the LDP passed a special measures bill to promote hunting and decentralization (policy core). |
| **Coalition Hypothesis 4 (CH4)** | Within a coalition, administrative agencies will usually advocate more moderate positions than their interest group allies. |
| | **0**: From Period 1–4, the Ministry of the Environment was a major actor promoting hunting. |
| **Coalition Hypothesis 5 (CH5)** | Actors within purposive groups are more constrained in their expression of beliefs and policy positions than actors from material groups. |
| | **2**: (e.g.,) While the left-wing parties, which can be considered as a material group that places importance on re-election through elections, began to approve of the promotion of hunting in Period 3, the nature conservation group, which is a special-interest group that places vlaue on wildlife protection, is a group that protects wildlife, and did not actively advocate for the promotion of hunting. |
| **Learning Hypothesis 1 (LH1)** | Policy-oriented learning across belief systems is most likely when there is an intermediate level of informed conflict between two coalitions. This requires that (1) each has the technical resources to engage in such a debate, and (2) the conflict is between secondary aspects of one belief system and core elements of the other or, alternatively, between important secondary aspects of the two belief systems. |
| | **2**: (e.g.,) In Period 4, policy-oriented learning that transcended belief systems took place. Since Period 1, many scientists had been part of the hunting coalition, and actors from the protection coalition had come to understand the severity of ungulate damage. When population estimates were reported at the Committee on Wildlife Protection and Management, it was shared that ungulate populations had grown excessively and that promoting hunting was necessary. As a result, both coalitions agreed on the need to promote hunting. The conflict that arose in Period 4 was limited to secondary beliefs, such as who would take on the responsibility of hunting. |
| **Learning Hypothesis 2 (LH2)** | Policy-oriented learning across belief systems is most likely when there exists a forum that is: (1) prestigious enough to force professionals from different coalitions to participate and (2) dominated by professional norms. |
| | **2**: (e.g.,) The expert meeting held during Period 4, The Committee on Wildlife Protection and Management, was a forum where scientific data played a central role, rather than a venue for political maneuvering. The population estimates for deer and the necessity of population control presented in this meeting facilitated policy-oriented learning that transcended belief systems, fostering a greater understanding of the need for hunting promotion even among members of the protection coalition. |
| **Learning Hypothesis 3 (LH3)** | Problems for which accepted quantitative data and theory that exist are more conducive to policy-oriented learning across belief systems than those in which data and theory are generally qualitative, quite subjective, or altogether lacking. |
| | **2**: (e.g.,) As demonstrated in LH2, the population estimates presented during Period 4 altered the attitudes of actors who had previously belonged to the protection coalition, leading them to recognize the necessity of promoting hunting. |
| **Learning Hypothesis 4 (LH4)** | Problems involving natural systems are more conducive to policy-oriented learning across belief systems than those involving purely social or political systems because in the former many of the critical variables are not themselves active strategists and because controlled experimentation is more feasible. |

*(Continued)*

**Table 1.** (Continued)

| Hypothesis | Hypotheses and their validation results |
|---|---|
| | **1**: (e.g.,) Although wildlife management is a matter related to natural systems, in Period 1 and Period 2, policy-oriented learning that transcended belief systems did not occur. However, as shown in LH2 and LH3, policy-oriented learning across belief systems occurred in Period 4. |
| **Learning Hypothesis 5 (LH5)** | Even when the accumulation of technical information does not change the views of the opposing coalition, it can have important impacts on policy—at least in the short run—by altering the views of policy brokers. |
| | **0**: I was unable to find any actors that could be considered policy brokers as envisioned by ACF. |
| **Policy Change Hypothesis 1 (PCH1)** | Significant perturbations external to the subsystem, a significant perturbation internal to the subsystem, policy-oriented learning, negotiated agreement, or some combination thereof is a necessary, but not sufficient, source of change in the policy core attributes of a governmental program. |
| | **2**: Internal and external perturbations, negotiated agreements and policy-oriented learning were observed in a series of major policy changes. |
| **Policy Change Hypothesis 2 (PCH2)** | The policy core attributes of a governmental program in a specific jurisdiction will not be significantly revised as long as the subsystem advocacy coalition that instated the program remains in power within that jurisdiction—except when the change is imposed by a hierarchically superior jurisdiction. |
| | **0**: The change of government, which was a change of the superior jurisdiction, had almost no effect on ungulate policy. |

Note: "2" indicates that the hypothesis was supported, "1" indicates that the hypothesis was partially supported, and "0" indicates that the hypothesis was rejected.

supported the legislation. It is suggested that the positioning of wildlife policy as an agroforestry policy (venue shopping) makes it difficult to oppose the promotion of hunting for damage control by material groups that seek support from many actors, such as political parties.

Second, while the ACF typically assumes that actors such as the ME, senior officials from the LDP, and the Cabinet act as neutral policy brokers, in this case, the ME and LDP played central roles within the hunting coalition. This deviation can be attributed to the strong influence of Japan's bureaucratic system and the LDP's dominance over the Cabinet. Many officials within the ME, responsible for managing national parks, were directly exposed to the escalating damage caused by deer populations and recognized the critical need for population control. Moreover, the LDP, whose political base includes rural areas heavily impacted by wildlife damage, had strong incentives to advocate for hunting policies. The party's close ties with the JHA reinforced this stance. Prominent LDP figures such as Toshihiro Nikai, Yoshihide Suga, and Shinzo Abe (who was Japan's longest-serving Prime Minister, holding office from 2006–2007 and 2012–2020) maintained direct connections with the JHA. Notably, during Period 1, Yohei Sasaki, an LDP politician, later became the chairman of the JHA, underscoring the close relationship between the LDP and JHA. Except for the brief period between 2009 and 2012, the LDP maintained control over the Cabinet and consistently supported hunting as a strategy for addressing ungulate damage. Consequently, potential policy brokers such as the Cabinet and the ME were not neutral but rather key actors within the hunting coalition.

Recent research on policy brokers has highlighted cases where candidates for policy brokers align with an advocacy coalition based on shared belief systems. (e.g., Sarvašová et al. [59]). In the Japanese context, Ohno et al. [33] argued that CH4 should be reconsidered due to the country's strong bureaucratic influence and the LDP's long-standing control over the Cabinet in Japan. This case study supports that view, indicating that policy brokers may not be as neutral as the ACF initially assumes.

Additionally, the concept of wildlife management, originally introduced by Aldo Leopold, is fundamentally based on rational, scientifically informed decision-making by administrative bodies and their staff [25]. Leopold's approach, emphasizing the role of science in wildlife management, has influenced modern practices in many countries, particularly in the United States (e.g., Organ et al. [60]). In Japan, the emphasis on scientific decision-making by administrative bodies is

similarly central to the wildlife management system [61]. This raises questions about the applicability of the ACF's CH4 hypothesis in contexts where scientific knowledge significantly shapes the policy beliefs of administrative agencies. Further research is needed to assess whether this phenomenon is unique to wildlife management or a broader characteristic of Japan's policy process."

## Conclusion

This paper examined the applicability of the Advocacy Coalition Framework (ACF) to wildlife management by analyzing policy changes in Japan's ungulate management. In response to the increasing damage caused by growing ungulate populations, a bottom-up policy change emerged, promoting hunting as a primary strategy. This shift was driven mainly by policy-oriented learning and was facilitated by decentralization—a factor originating outside the wildlife management policy subsystem. These elements collectively expanded the resources available to the hunting coalition, ultimately shaping Japan's current approach to managing ungulates.

The study's findings underscore the value of applying the ACF to wildlife management. The framework enables a systematic understanding of the complex dynamics at play, moving beyond the ad hoc interpretations typical of individual case studies. By employing the ACF, this research has provided a structured analysis of the interactions between advocacy coalitions, policy belief, policy-oriented learning, internal and external perturbation, and policy change. Additionally, within the context of ACF research, this study suggests the need for a reexamination of hypotheses regarding policy brokers and emphasizes the importance of venue shopping and policy feedback. It demonstrates how studies on wildlife management can contribute to the development of the ACF.

Looking forward, further research applying the ACF to wildlife management across different contexts—such as at the other countries or prefectural or municipal levels, or involving different wildlife species—will enrich our understanding of wildlife management as an academic field. Expanding ACF-based wildlife management studies will help reveal broader patterns and dynamics, contributing to more comprehensive and effective wildlife management strategies across diverse settings.

## Supporting information

**S1 Table. The ACF hypotheses.**
(DOCX)

**S2 Table. Interview guide.**
(DOCX)

**S3 Table. List of organizations in policy subsystem.**
(DOCX)

**S4 Table. Coding scheme.**
(DOCX)

## Acknowledgments

I received help from Sako Uematsu (Tokyo University of Agriculture and Technology) regarding the verification of coding.

## Author contributions

**Conceptualization:** Tatsuya Koga.

**Data curation:** Tatsuya Koga.

Formal analysis: Tatsuya Koga.

Investigation: Tatsuya Koga.

Methodology: Tatsuya Koga.

Project administration: Tatsuya Koga.

Resources: Tatsuya Koga.

Validation: Tatsuya Koga.

Visualization: Tatsuya Koga.

Writing – original draft: Tatsuya Koga.

Writing – review & editing: Tatsuya Koga.

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
