## [Decision Letter · Decision Letter 0]

14 Jul 2025

PONE-D-25-19409Applying the Advocacy Coalition Framework to Wildlife Management: Explaining Policy Change for Damage Mitigation in JapanPLOS ONE

Dear Dr. Koga,

Thank you for submitting your manuscript to PLOS ONE. After careful consideration, we feel that it has merit but does not fully meet PLOS ONE’s publication criteria as it currently stands. Therefore, we invite you to submit a revised version of the manuscript that addresses the points raised during the review process.

Considering the reviewers' suggestions, I request that the authors make the necessary adjustments (these should be highlighted in a different color for identification) and return the paper so we can proceed with publication. 

Best regards

We look forward to receiving your revised manuscript.

Kind regards,

Julio Cesar de Souza, Ph.D.

Academic Editor

PLOS ONE

Journal Requirements:

2. You indicated that ethical approval was not necessary for your study. We understand that the framework for ethical oversight requirements for studies of this type may differ depending on the setting and we would appreciate some further clarification regarding your research. Could you please provide further details on why your study is exempt from the need for approval and confirmation from your institutional review board or research ethics committee (e.g., in the form of a letter or email correspondence) that ethics review was not necessary for this study? Please include a copy of the correspondence as an ""Other"" file.

3. Thank you for stating the following in the Competing Interests section* (delete as necessary):

“The Forestry and Forest Products Research Institute (FFPRI),　to which I belong, is an organization that appeared in the policy subsystem in Periods 1 and 2, but I declare that there are no known competing financial interests or personal relationships that could affect the research reported in this paper.”

We note that one or more of the authors have an affiliation to the commercial funders of this research study : The Forestry and Forest Products Research Institute (FFPRI)

5. In the online submission form, you indicated that [Insert text from online submission form here].

Additional Editor Comments (if provided):

Considering the reviewers' suggestions, I request that the authors make the necessary adjustments (these should be highlighted in a different color for identification) and return the paper so we can proceed with publication.

Best regards

Reviewers' comments:

Reviewer's Responses to Questions

**Comments to the Author**

1. Is the manuscript technically sound, and do the data support the conclusions?

Reviewer #1: Partly

Reviewer #2: Partly

Reviewer #3: Yes

2. Has the statistical analysis been performed appropriately and rigorously? 

Reviewer #1: Yes

Reviewer #2: No

Reviewer #3: Yes

3. Have the authors made all data underlying the findings in their manuscript fully available?

Reviewer #1: Yes

Reviewer #2: Yes

Reviewer #3: Yes

4. Is the manuscript presented in an intelligible fashion and written in standard English?

Reviewer #1: Yes

Reviewer #2: Yes

Reviewer #3: Yes

5. Review Comments to the Author

Reviewer #1: Based on the methods, the authors should give on the Table S3 the number of person in each institution asked. It might be a soiurce of biais the fact, that there is no balance on this number. Example, the number of Prefectural Government are at least represetned by five entities, but the Hunting group is only one. So youshould give explaination on how to avoid "biais" on getting more response trends towards governemental perception than those of hunters

Reviewer #2: This is an interesting study author tried to compile through the article. However, it is suggested to revisit the manuscript and consider it in submitting review or essay section. Yes data has been analyzed through different source and interviews though it is not reflecting the results and further in discussion and conclusion section. At current stage manuscript is very descriptive which suits for review or essay

Reviewer #3: Thank you for the opportunity to review this manuscript. Overall, I found it to be well-written, clearly structured, and scientifically sound. The research question is well-motivated, and the methods and conclusions are appropriate for the scope of the journal.

I have no major concerns with the content. However, I do have a minor suggestion:

Figure Clarity: Some of the figures (e.g., Figures 1 and 2) are difficult to read due to low resolution or small text labels. Improving the clarity and resolution would enhance readability and overall presentation quality.

Otherwise, I believe the manuscript is in good shape and can proceed with minor revisions.

6. PLOS authors have the option to publish the peer review history of their article (what does this mean? ). If published, this will include your full peer review and any attached files.

**Do you want your identity to be public for this peer review?** For information about this choice, including consent withdrawal, please see our Privacy Policy .

Reviewer #1: **Yes: ** Aristide Andrianarimisa

Reviewer #2: No

Reviewer #3: No

---

## [Author Response · Author response to Decision Letter 1]

18 Aug 2025

Dear Reviewer#1

Thank you very much for your insightful comment regarding the potential for sampling bias.

In response to your concern, I have revised the manuscript to clarify the following two points, demonstrating that no significant sampling bias occurred in this study.

1. Response to Concerns Regarding Interviewee Distribution

I appreciate your suggestion and have added the number of interviewees per institution in Table S3.

I acknowledge the imbalance in the number of interviewees across different types of organizations. Unfortunately, some key individuals had retired or passed away, which made it difficult to reach all relevant stakeholders. This limitation is now explicitly explained in lines 326–330 of the main text.

However, as described in the main text, the interviews mainly served to supplement the core data sources. The primary analysis of the policy process and discourse network construction relied on publicly available primary sources, including newspaper articles, parliamentary minutes, and official government documents. Therefore, I believe that the identification and analysis of key policy actors and their belief systems were conducted comprehensively and without bias.

2. Clarification Regarding Hunting Organizations

I appreciate your comment on the representation of hunters in Table S3. Indeed, only one hunting organization—the Japan Hunters’ Association (JHA)—was included, while multiple prefectural governments were listed.

To clarify, hunters in Japan are organized at the local level into branch clubs coordinated by prefectural hunting associations. All of these fall under the umbrella of the JHA, which is the sole national-level organization representing hunters in policy discussions.

Although other smaller hunters’ groups exist, they have limited political influence and do not engage in national policy-making processes. The JHA is the only hunters’ organization regularly participating in national-level decision-making bodies, such as government advisory councils.

For this reason, I considered the JHA to be the sole representative of hunters at the national level—comparable to how individual prefectural governments are included based on their direct involvement in national policy processes.

To clarify this for readers, I propose adding a footnote to Table S3 explaining the organizational structure of hunters’ associations in Japan and the rationale for focusing on the JHA. I hope this addition will demonstrate that my analytical focus reflects the actual distribution of political influence rather than sampling bias.

Thank you again for your valuable feedback.

Best regards,

Tatsuya Koga

Dear Reviewer #2,

Thank you very much for your thoughtful feedback.

While I understand your concern regarding the descriptive nature of the Results section, I would like to clarify that this manuscript primarily employs a qualitative policy process analysis grounded in the Advocacy Coalition Framework (ACF). Within this tradition, detailed narrative accounts are often necessary to trace the evolution of policy beliefs, actor coalitions, and institutional shifts.

That said, I have carefully reviewed the Results section and made revisions to make the analytical structure more explicit. Specifically, I clarified how the evidence supports changes in belief systems, coalition alignments, and policy outcomes over time. I hope these revisions help demonstrate the analytical depth of the study.

Major Revisions:

In the Results section, I revised the structure by introducing or modifying subheadings for each period and reorganizing the order of paragraphs. By doing so, I aimed to make the causal pathways—from policy beliefs, coalition structures, and policy-oriented learning to external perturbations and eventual policy change—more accessible and comprehensible to readers from diverse disciplinary backgrounds.

Period 1: Intensifying Deer Damage → Coalition Structures and Policy-Oriented Learning → External Perturbations and Policy Change

Period 2: Policy feedback of decentralization in Period 1, Coalition Structure and Policy-Oriented Learning → Policy Change Toward Relaxation of Hunting Regulations

Period 3: Venue Shopping → Coalition Structure and Policy Change

Period 4: Change in Government → Policy-Oriented Learning → Coalition Structure and Policy Change

In addition, I revised wording and reorganized sentence order throughout the manuscript in accordance with the reviewers’ comments.

Best regards,

Tatsuya Koga

Dear Reviewer #3,

Thank you very much for reviewing my manuscript and for your positive evaluation.

In response to your comment regarding the resolution of the figures, I have improved their quality accordingly.

Thank you again for your valuable feedback.

Sincerely,

Tatsuya Koga

---

## [Editor Report · Decision Letter 1]

24 Aug 2025

Applying the advocacy coalition framework to wildlife management: explaining policy change for damage mitigation in Japan

PONE-D-25-19409R1

Dear Dr. Koga,

We’re pleased to inform you that your manuscript has been judged scientifically suitable for publication and will be formally accepted for publication once it meets all outstanding technical requirements.

Kind regards,

Julio Cesar de Souza, Ph.D.

Academic Editor

PLOS ONE

Additional Editor Comments (optional):

Considering that there were minor suggestions and that the authors made the necessary adjustments, which they considered prudent,

I am in favor of publication.

JCS
---

## [Editor Report · Acceptance letter]

PONE-D-25-19409R1

PLOS ONE

Dear Dr. Koga,

I'm pleased to inform you that your manuscript has been deemed suitable for publication in PLOS ONE. Congratulations! Your manuscript is now being handed over to our production team.

Kind regards,

on behalf of

Dr. Julio Cesar de Souza

Academic Editor

PLOS ONE